

# Improved plaque assay for human coronaviruses 229E and OC43

Nicole Bracci[1,*], Han-Chi Pan[2,*], Caitlin Lehman[1], Kylene Kehn-Hall[1] and Shih-Chao Lin[3]

[1] Department of Biomedical Sciences and Pathobiology, Virginia-Maryland College of Veterinary Medicine, Virginia Polytechnic Institute and State University (Virginia Tech), Blacksburg, VA, USA
[2] National Laboratory Animal Center, National Applied Research Laboratories, Taipei, Taiwan
[3] Bachelor Degree Program in Marine Biotechnology, National Taiwan Ocean University, Keelung, Taiwan
* These authors contributed equally to this work.

Corresponding authors
Kylene Kehn-Hall, kkehnhall@vt.edu
Shih-Chao Lin, sclin@mail.ntou.edu.tw

## ABSTRACT

In light of the COVID-19 pandemic, studies that work to understand SARS-CoV-2 are urgently needed. In turn, the less severe human coronaviruses such as HCoV-229E and OC43 are drawing newfound attention. These less severe coronaviruses can be used as a model to facilitate our understanding of the host immune response to coronavirus infection. SARS-CoV-2 must be handled under biosafety level 3 (BSL-3) conditions. Therefore, HCoV-229E and OC43, which can be handled at BSL-2 provide an alternative to SARS-CoV-2 for preclinical screening and designing of antivirals. However, to date, there is no published effective and efficient method to titrate HCoVs other than expensive indirect immunostaining. Here we present an improved approach using an agarose-based conventional plaque assay to titrate HCoV 229E and OC43 with mink lung epithelial cells, Mv1Lu. Our results indicate that titration of HCoV 229E and OC43 with Mv1Lu is consistent and reproducible. The titers produced are also comparable to those produced using human rhabdomyosarcoma (RD) cells. More importantly, Mv1Lu cells display a higher tolerance for cell-cell contact stress, decreased temperature sensitivity, and a faster growth rate. We believe that our improved low-cost plaque assay can serve as an easy tool for researchers conducting HCoV research.

## INTRODUCTION

The coronavirus disease 2019 (COVID-19) pandemic, caused by severe acute respiratory syndrome coronavirus 2 (SARS-CoV-2), has dramatically altered the way of life worldwide and halted non-essential scientific research. As a result, the priority of the world is to develop antiviral agents and vaccines against SARS-CoV-2 (*Wu et al., 2020*). Despite the catastrophic effects brought on by SARS-CoV-2, other human coronaviruses (HCoV), such as the strains 229E and OC43, have been circulating for years and are one of the causative agents for the common cold (*Greenberg, 2016*). Due to the low lethality of non-severe coronavirus strains, they tend to be neglected, and resources to investigate their pathogenesis and epidemiology are relatively limited (*Zeng et al., 2018*; *Zumla et al., 2016*).

However, the outbreaks of SARS in 2003, Middle East respiratory syndrome (MERS) in 2012, and COVID-19 in 2019-current, have drawn an unprecedented amount of attention to coronavirus research, yet there is no effective method to perform a low-cost plaque assay for HCoV 229E and OC43. Being able to titrate these strains accurately would facilitate a plethora of coronavirus studies.

Due to the lack of effective antivirals and vaccines towards the highly contagious coronavirus strains, such as SARS-CoV, MERS-CoV, and SARS-CoV-2, it is required to work with these agents in a negative pressure equipped biosafety level 3 (BSL-3) laboratory or above. In addition to work in these facilities, personnel have to be well-trained and wear proper personal protection equipment (PPE), including Tyvek suits and respiratory protection such as powered air-purifying respirators. However, a great number of institutes lack the budget or even the capability to perform work in such a facility. In addition, PPE supplies have been in high demand during the COVID-19 pandemic, limiting researchers' access to PPE. As a result, studies focusing on these non-severe coronavirus strains have become a necessary alternative to prescreen potential antiviral compounds in a BSL-2 setting. Utilizing these strains allows for a readily available low-cost alternative that provides insight into the viral properties of the human coronaviruses.

While RT-qPCR has been widely adapted to determine the viral loads of CoVs due to its sensitivity, particularly for SARS-CoV-2; it is also very costly for the required reagents for researchers who lack the budget. Additionally, RT-qPCR is incapable of reflecting virucidal activity of a given drug if it targets infectious virion production or egress instead of viral genome replication. As a result, a conventional plaque assay can be a viable low-cost and complement approach for RT-qPCR to test a potential antiviral compound and quantify infectious viral titers as an indication of viral loads in terms of virological research.

Unlike non-severe human CoVs, SARS- and MERS-CoVs can easily form plaques using Vero cells (*Coleman & Frieman, 2015*; *Harcourt et al., 2020*; *Vicenzi et al., 2004*). Vero cells are a non-cancerous cell line with a type I interferon production defect (*Emeny & Morgan, 1979*) and are the most commonly used cell line for the traditional plaque assays (*Baer & Kehn-Hall, 2014*). However, the traditional plaque assay appears to be inadequate to titrate HCoV 229E and OC43. Without an accurate titration, it would be difficult to estimate drug efficacy and perform vaccine challenge studies. However, there are alternative approaches to titrate HCoV 229E and OC43. For example, an indirect plaque assay using an anti-CoV primary antibody and peroxidase to visualize the virus-infected cell colonies has been developed (*Lambert et al., 2008*). Direct observation of cytopathogenic effects (CPE) by the naked eye utilizing a compound light microscope can even be performed to determine the TCID50 (Median Tissue Culture Infectious Dose). However, the CPE-based titration is not sensitive enough to discern viral inhibition resulting from potential antiviral compounds, and the indirect immunoperoxidase assay consumes large amounts of antibodies, which makes this method expensive despite its high sensitivity.

Approaches to improve the reliability of the traditional plaque assay include utilizing more susceptible cell lines and replacing the overlay medium with cellulose materials such

as low-viscosity Avicel or methylcellulose (*Funk et al., 2012*; *Matrosovich et al., 2006*). For example, *Herzog, Drosten & Muller (2008)* reported that while the LLC-MK2 cells, a rhesus monkey kidney cell line, is suitable for propagation of HCoV-NL63, one of the non-severe strains isolated in 2007 (*Pyrc, Berkhout & Van der Hoek, 2007*), it is not suitable to titrate NL63 using conventional plaque assay due to no or little CPE observed. A similar result was also observed with Vero cells, following NL63 infection. To combat this, their study used Caco-2 cells as a monolayer and applied Avicel RC-581 as the overlay medium to successfully titrate NL63 by traditional plaque assay (*Herzog, Drosten & Muller, 2008*). Despite their success, very limited information regarding the conventional plaque assays for HCoV-229E and OC43 can be found. As a result, this study revisited the traditional plaque assay method for HCoV 229E and OC43 and validated the proposed protocols. Human lung epithelial cells, MRC-5 and ileocecal adenocarcinoma cells, HTC-8, were used to propagate 229E and OC43, respectively (*Gorse et al., 2009*; *Laude & Vautherot, 1993*; *Warnes, Little & Keevil, 2015*). Various cell lines were then utilized to optimize an agarose-based conventional plaque assay. The goal of this study was to create a plaque assay protocol that could stably and reproducibly titrate the non-severe coronavirus strains HCoV 229E and OC43.

## MATERIALS AND METHODS

### Cells, viruses, and medium

The cell lines used in this study were purchased from ATCC: MRC-5 (human lung epithelial cell; ATCC: CCL-171), HCT-8 (human ileocecal adenocarcinoma cell; ATCC: CCL-244), Mv1Lu (mink lung epithelial cell; ATCC: CCL-64), RD (human rhabdomyosarcoma cell; ATCC: CCL-136), LLC-MK2 (rhesus monkey kidney cell; ATCC: CCL-7), Vero (African green monkey kidney cell; ATCC: CCL-81). HCoV-229E (Catalog No. FR-303) and OC43 (Catalog No. FR-302) were obtained from International Reagent Resources (IRR) followed by propagation in MRC-5 and HCT-8 cells for 3–5 days post infection, respectively.

Medium for MRC-5, Mv1Lu, RD, LLC-MK2, and Vero cells was DMEM (Cat# 112-300; Quality Biological, Gaithersburg, MD, USA) supplemented with 10% FBS, 1× MEM nonessential amino acids (NEAA; Cat# 25-025; Corning, Corning, NY, USA), 1 mM of sodium pyruvate (Cat# 25-000; Corning, Corning, NY, USA), 2 mM of L-glutamine (Cat# 25-005; Corning, Corning, NY, USA), and 1× penicillin and streptomycin (Cat#30-002; Corning, Corning, NY, USA) except HCT-8 which was cultured with RPMI 1640 (Cat# 112-025; Quality Biological, Gaithersburg, MD, USA) containing 10% FBS.

### Plaque assay

A total of $1 \times 10^6$ cells per well were seeded in a 6-well plate a day before performing the plaque assay. The culture media was removed and washed with PBS. Viral samples were diluted 10-fold in DMEM without FBS and the appropriate dilution was added to the corresponding well and incubated for 1.5 h at 37 °C and 5% $CO_2$. Overlay medium was prepared from a mixture of 1% of agarose (Cat#16500; Invitrogen, Carlsbad, CA, USA), unless indicated elsewhere, and 2× EMEM (Cat# 115-073; Quality Biological,

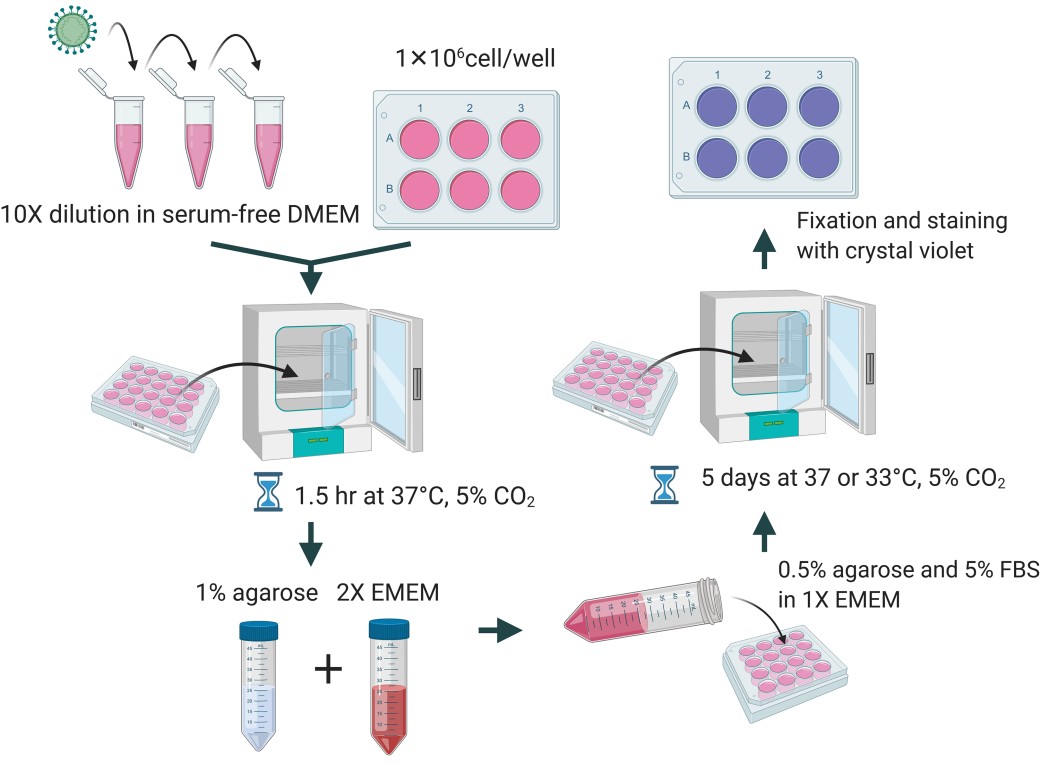

**Figure 1 Diagrammatic scheme for illustration of improved plaque assay procedures and optimal conditions.** Figure created with https://Biorender.com.

Gaithersburg, MD, USA) supplemented with 10% or 16% of FBS, 1× NEAA, 1 mM of sodium pyruvate, 2 mM of L-glutamine, and 2× penicillin and streptomycin. Two mL of the overlay medium were added to each well. The plates were incubated at 37 °C or 33 °C and 5% $CO_2$ for 5 days. At the end of incubation, cells were fixed with 10% formalin for at least 1 h and stained with 0.2% crystal violet dissolved in 10% of absolute ethanol and 90% of PBS overnight. A diagrammatic scheme is depicted in Fig. 1.

# RESULTS

## Plaque assays of HCoV 229E and OC43 with MRC-5 cells

MRC-5 cells have successfully been used to titrate HCoV 229E and OC43 in previous studies (*Funk et al., 2012*; *Kim et al., 2019*; *Warnes, Little & Keevil, 2015*). This study attempted to reproducibly test whether MRC-5 cells may be used to titrate HCoV 229E and OC43 using the proposed agarose plaque assay method. The ability of each strain to form plaques was analyzed. Additionally, it was examined if the presence or absence of the inoculum in the wells had any effect on plaque formation. It was found that when the inoculum remained in the wells throughout the 5-day incubation period, the monolayer of the MRC-5 cells appeared to be intact and conducive to plaque formation. Conversely, without the inoculum, MRC-5 cells hardly remained intact and seemed unable to support the formation of viral plaques (Fig. 2). Various compositions of overlay medium were

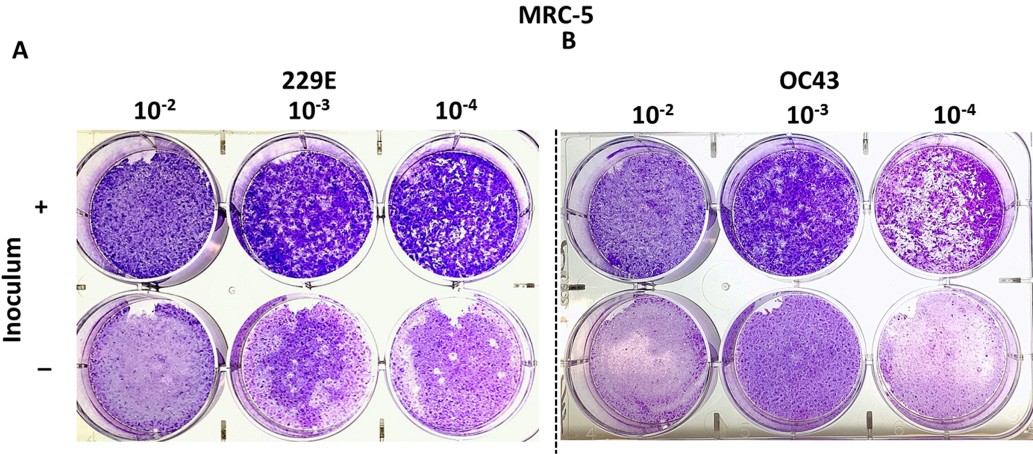

**Figure 2 Plaque assays with MRC-5 cells.** Plaque assay for (A) 229E and (B) OC43 on MRC-5 monolayer with or without inoculum left in wells. The inoculum was incubated for 1.5 h at 37 °C. Overlay medium was composed of a final conc. of agarose ~0.75% and ~8% of FBS.

tested by adjusting the concentrations of agarose and FBS. An optimal concentration of 0.75% agarose and 8% FBS stood out among the various overlay mediums tested since it formed the most discernable plaques on MRC-5 cells (Fig. S1).

## Optimize agarose-based plaque assays for HCoV 229E and OC43

Although countable plaques were formed in MRC-5 cells, we realized that MRC-5 cells are not an ideal cellular material for plaquing HCoV 229E and OC43. MRC-5 cells are slow-growing and are very sensitive to their surrounding growth conditions, which are not ideal for the formation of a monolayer with a 5-day incubation. Therefore, other cell types were compared to the MRC-5 cells for their susceptibility to HCoV 229E and OC43 by the observance of cytopathic effects (CPE). Among the cell types tested, including LLC-MK2, which has been previously described as permissive for SARS-CoV (*Kaye, 2006*) and NL-63 (*Schildgen et al., 2006*), the mink epithelial cells, Mv1Lu, initially used in the detection of various respiratory viruses (*Huang & Turchek, 2000*), were selected due to their significant CPE at 4 days post HCoV 229E and OC43 infection (Fig. 3). As a result, Mv1Lu cells were used to optimize the plaque assay protocol.

For coronaviruses as well as many other respiratory viruses, 33 °C appears to be the optimal temperature for propagation and titration (*Creager et al., 2018*; *Lamarre & Talbot, 1989*; *Muller et al., 2017*). To confirm this, plaque assays were performed with Mv1Lu cells at both 33 °C and 37 °C, and the plaques formed were compared. Following the 5 day incubation period, it was found that the plaque sizes of HCoV 229E and OC43 were smaller at 33 °C than those at 37 °Furthermore, the titer of OC43 was significantly reduced at 33 °C while there was no notable titer change for 229E at either temperature (Figs. 4A and 4B). These results indicate that 37 °C appears to be an appropriate temperature for the modified plaque assay with Mv1Lu cells.
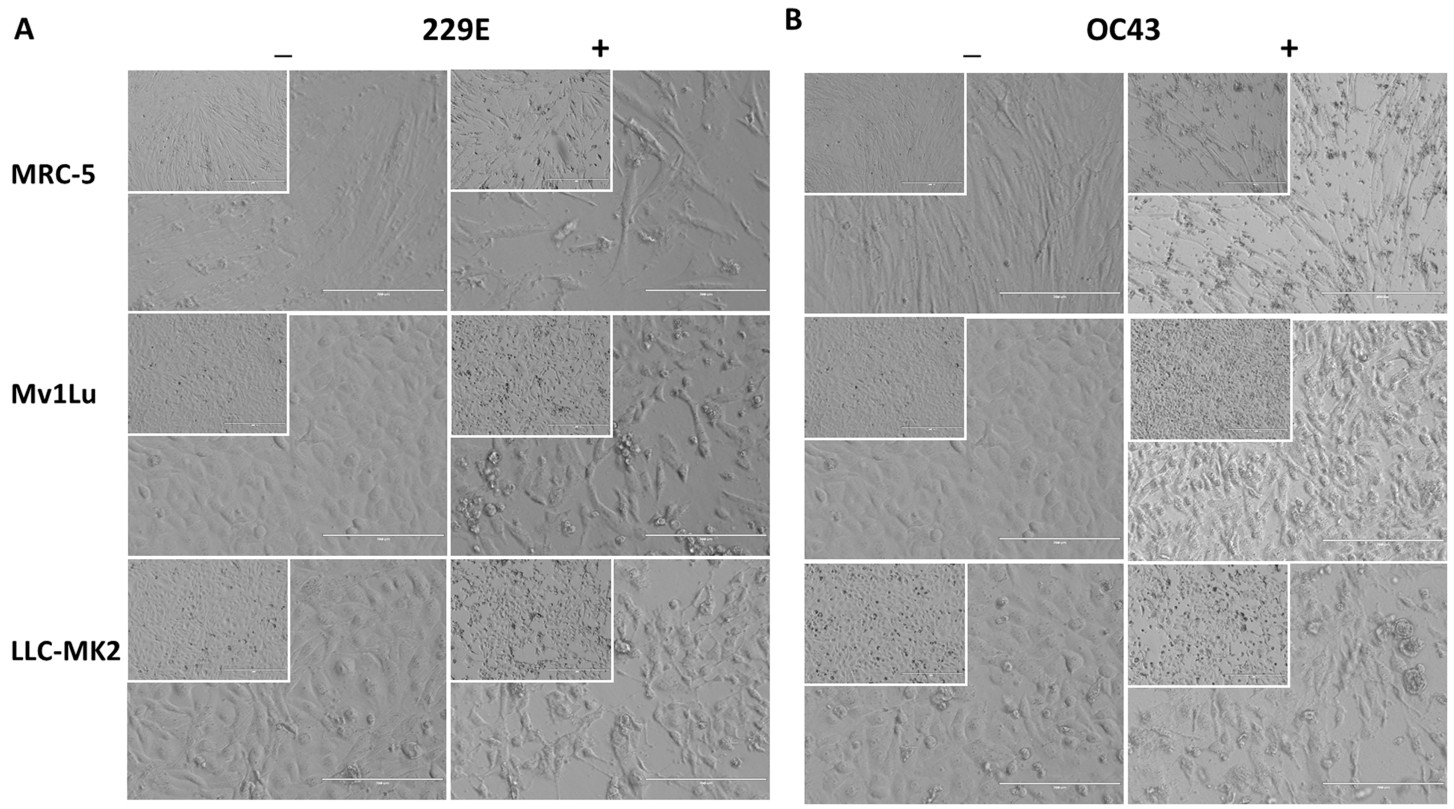

**Figure 3 Cell morphology of susceptible cell lines.** MRC-5, Mv1Lu, and LLC-MK2 with or without infections of (A) 229E and (B) OC43 at 4 dpi. Magnification 200×; Insert 100×.

Next, various agarose concentrations were tested to determine which concentration formed the most discernable plaques (*Shurtleff et al., 2012*). The results showed no obvious difference in shape, size, or number of plaques generated with either HCoV 229E or OC43 at any of the final agarose concentrations of 1%, 0.5%, or 0.3% (Fig. 4C). The final concentration of 0.5% agarose was used in the next sequence of experiments due to its ease in handling and the clarity of plaques formed. Therefore, the optimal conditions for the plaque assay protocol for 229E and OC43 utilize Mv1Lu cells, incubated at 37 °C with a final agarose concentration of 0.5%.

## Validation of agarose-based plaque assays with longitudinal viral passages

To validate the proposed plaque assay protocol, two consecutive passages of HCoV 229E and OC43 from MRC-5 and HCT-8 were cultured and collected, respectively. The corresponding titers (Fig. 5A) and the plaque morphologies (Figs. 5B and 5C) were noted. The plaque assay conditions used, including temperature and overlay medium composition were adequately able to discern the differences in viral titers of HCoV 229E and OC43 between two passages. Both strains were at least 10-fold greater after passaging. The data demonstrates that the current modified plaque assay protocol with Mv1Lu cell can reproducibly differentiate a change in titer of HCoVs.

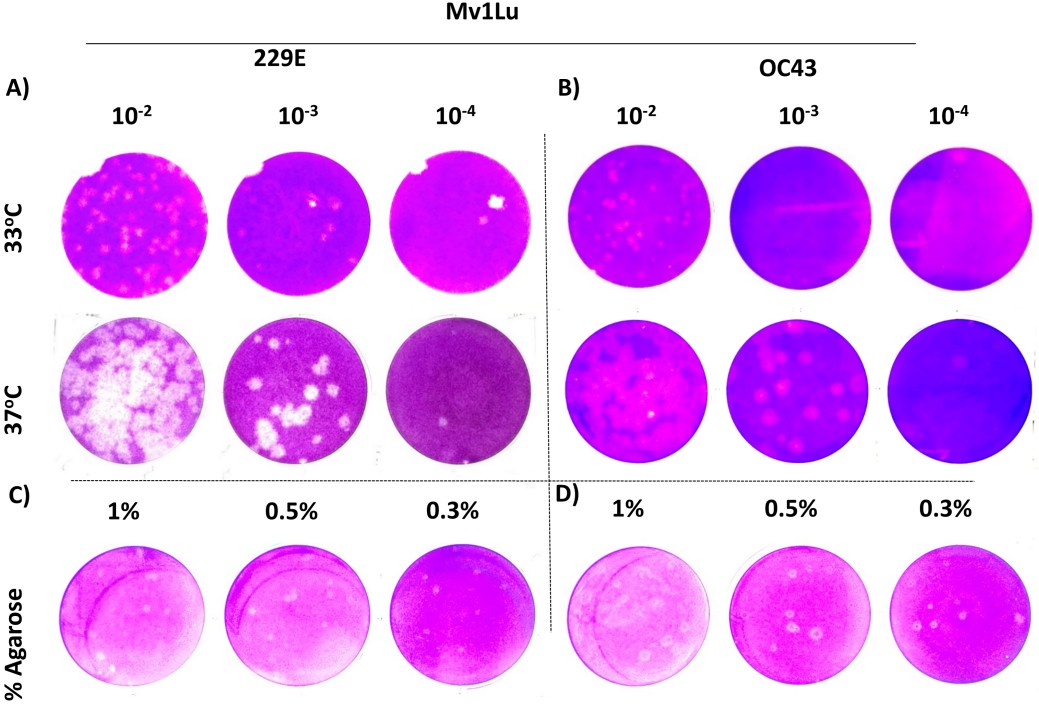

**Figure 4 A tune-up for plaque assay conditions of hCoVs.** Plaques observed at 5 dpi of (A) 229E and (B) OC43 on Mv1Lu monolayer at 33 °C and 37 °C with overlay media of 0.5% agarose and 5% FBS. A representative sample of plaques formed by (C) 229E and (D) OC43 using various overlay mediums where indicated final agarose concentrations were mixed with 2X EMEM media supplemented with 10% FBS and other ingredients described in "Material and Methods".

## Comparison of Mv1Lu and RD cells for HCoV 229E and OC43 plaque assays

Human rhabdomyosarcoma (RD) cells have also been reported as a conducive cell line for HCoV- 229E and OC43 plaque formation (*Schmidt, Cooney & Kenny, 1979*), but over the years MRC-5 cells have also become common cell lines for HCoVs plaque assays. To further characterize these cell types' ability to form plaques, the optimized protocol utilizing both cell lines was performed concurrently. Following the 5-day incubation, both cell lines exhibited a decent capability in generating countable and discernable plaques (Fig. 6). Overall, both cell lines are adequate as cellular material to titrate HCoV 229E and OC43 with slightly different properties in forming clear plaques. That is, the monolayer of Mv1Lu cells was easily stained by the crystal violet solution which yielded clearer and less hazy plaques than those on RD cells. Nevertheless, both Mv1Lu and RD cells exhibited adequate and comparable capability in titrating 229E and OC43.

## Comparison of HCoV 229E and OC43 propagated from different cell lines

Lastly, given that a virus can be propagated in cell lines that may not be permissive for plaque formation, three cell lines, MRC-5 (human lung epithelial cell), Mv1Lu (mink lung epithelial cell), and LLC-MK2 (rhesus monkey kidney epithelial cell) cells, were
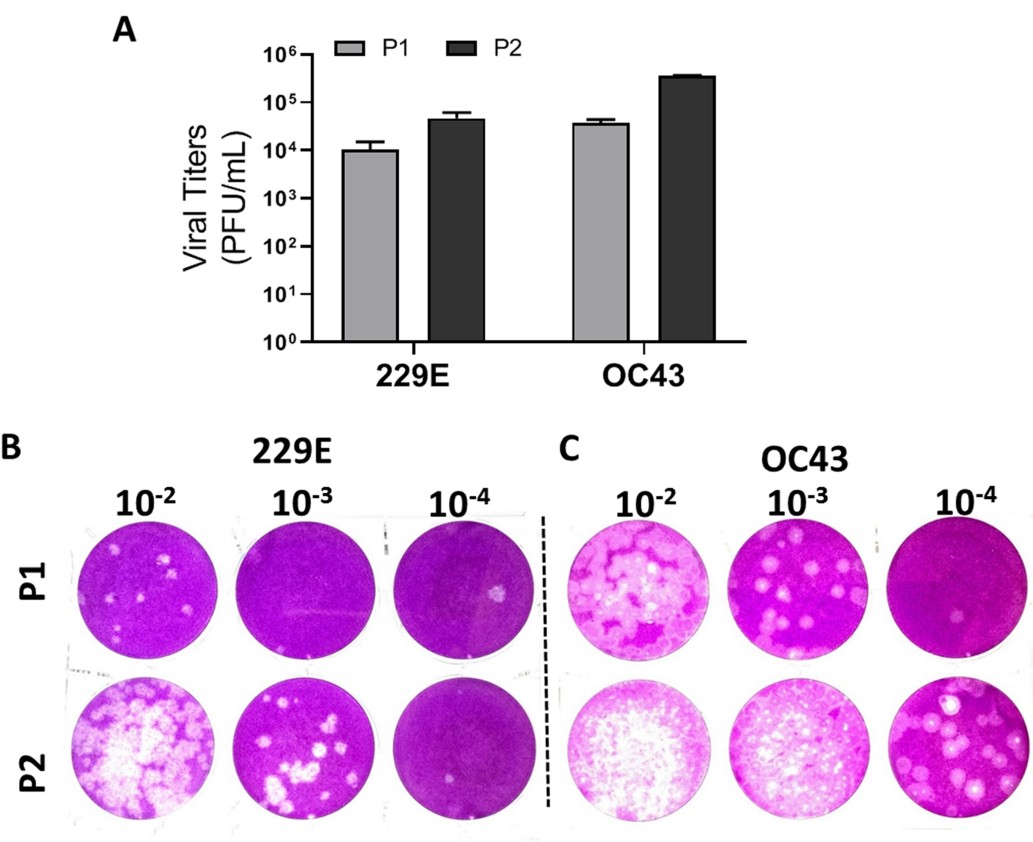

**Figure 5 Functional evaluation with improved plaque assay.** Titration of consecutive passages of (A and B) 229E and (A and C) OC43 by plaque assay at 37 °C with overlay media of 0.5% agarose and 5% FBS.

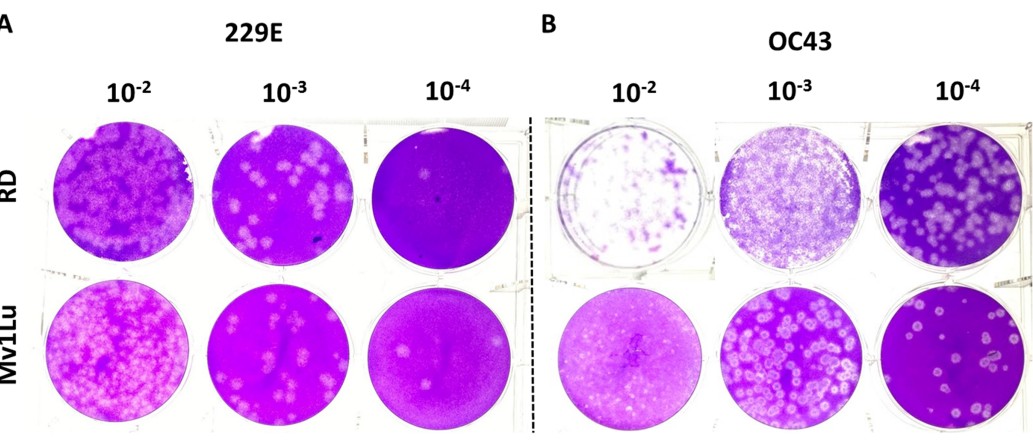

**Figure 6 Comparison of susceptible cell lines with the improved plaque assay.** Plaques formed at 5 dpi by (A) 229E and (B) OC43 using a Mv1Lu and RD cell monolayer using an EMEM-based overlay media of 0.5% of agarose and 5% of FBS.

screened for their ability to propagate HCoV 229E and OC43. Viral titers yielded from the 3 distinct cell lines were determined utilizing the proposed modified plaque assay method. Results showed that the viral titers of HCoV 229E and OC43 generated from MRC-5 were

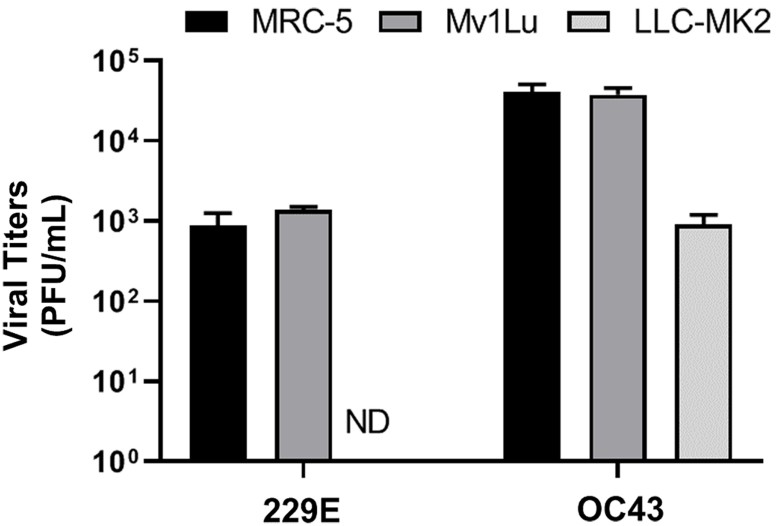

**Figure 7 Titration of hCOV propagation from different cells.** Differential titers of 229E and OC43 propagated from three distinct cell lines. Virus-containing supernatants were collected on 4 dpi and titrated by using the 5 day plaque assays protocol. Values are presented as mean ± SEM from at least two independent experiments.                                      

comparable to those from Mv1Lu cells. Whereas the titer obtained for OC43 propagated in LLC-MK2 cells was down by nearly 2 logs and no virus could be detected for 229E propagated in LLC-MK2 cells (Fig. 7). This experiment was then repeated using LLC-MK2 as the monolayer for the modified plaque assay at 37 °C and 33 °C. Our results indicate that LLC-MK2 cells showed a limited capability to support plaque formation of HCoV OC43 and did not support plaque formation of HCoV 229E at either temperature (Fig. S2). The plaque assays results correspond with the inability of HCoV 229E to replicate in LLC-MK2.

## DISCUSSION

In this study, we demonstrated that HCoV 229E and OC43 can be reproducibly titrated by a modified conventional plaque assay protocol using Mv1Lu cells. It was also shown that the use of Mv1Lu cells produces plaques that are comparable and even in cases superior to those of RD cells. Mv1Lu cells are derived from American mink lung epithelial cells and they have been adapted to support propagation of various viruses, specifically influenza virus and respiratory syncytial virus (*Schultz-Cherry et al., 1998*; *Tsai et al., 2019*; *Yeolekar et al., 2002*). From our observations, Mv1Lu cells seem to grow faster (subculture frequency on average of 3 days) and are more tolerant of cell-cell contact stress. It was also observed that Mv1Lu cells in a culture flask remain morphological intact for at least 7–9 days without the addition of new culture medium or subculturing. Whereas MRC-5 cells grow relatively slow (subculture frequency on average of 5 days; Fig. S3) and require more meticulous culture conditions to prevent monolayer breakdown during incubation. RD cells are another adequate cell line to support plaque formation of HCoV 229E and OC43. However, the morphology of the plaques formed in RD cells is not as defined as those formed in Mv1Lu cells for OC43 (Fig. 6B). Vero cells were also tested

for their ability to form plaques upon infection with HCoV at 33 °C with a similar but distinct protocol (see "Supplemental Protocol"). Unfortunately, Vero cells were not supportive of plaque formation for either strain (data not shown). An altered protocol might enable plaque formation in Vero cells since extensive optimization of this protocol was not performed.

Agarose is a widely used viscous material found in many overlay mediums. It is efficient at restraining diluted viral particles which aid in the formation of distinct countable plaques. Avicel and methylcellulose are other widely used materials for overlay medium. Avicel is a nonfibrous, microcrystalline cellulose derivative commonly used by the food industry, one example of its use is in ice cream production (*Kennedy & Law, 1999*). The use of Avicel, as a low viscosity overlay medium, provides optimal conditions for indirect immunoperoxidase assays which is ideal for those viruses that don't readily form plaques like influenza (*Lin et al., 2017*; *Matrosovich et al., 2006*). Previously, Avicel has been reported to be an outstanding overlay medium for titrating another HCoV strain, NL63, compared to agarose and methylcellulose (*Herzog, Drosten & Muller, 2008*). However, Avicel powder is not easily acquired and the preparation of a homogenous solution is more complicated and time-consuming than agarose. In this study plaques of NL63 could not be visualized with Mv1Lu or RD cells using an agarose-based protocol (data not shown).

It was observed that MRC-5 is not the ideal cell type for conducting our modified plaque assay for HCoVs due to the following reasons: (1) inconsistent plaque formation of HCoVs; (2) The cell line possesses a narrow window of observation to obtain clear and readable plaques; and (3) MRC-5 cells have a higher demand for culture conditions. In a 10-fold dilution of HCoV 229E and OC43 samples, the countable plaques obtained in a 6-well plate are present in only one single dilution. This means a 12-well plate format is not recommended for plaque assays performed with MRC-5 cells because this format yields an insufficient resolution for counting plaques. Discernable plaques were not formed with the HCoV 229E in MRC-5 cells until 4 to 5 days post incubation, which differs from the previously described 48 h protocol (*Funk et al., 2012*). In our experience, a 4-day incubation with Mv1Lu or RD cells is also a viable option to yield countable plaques. With MRC-5 cells, even an incubation as short as 4 days requires a higher concentration of nutrients (8% of FBS) compared to the other cell lines used in this study (2.5–5% of FBS). The aforementioned factors would substantially increase the variations and costs due to the increased labor, supplies, and reagents needed to perform large-scale anti-HCoV drug testing. However, if it is necessary to perform plaque assays on MRC-5 cells, using an alternative overlay such as 0.8% of methylcellulose may allow for a higher resolution.

It is important to note that cell lines that support plaque formation are not necessarily the ideal candidate for viral stock propagation. For example, it has been reported that HCoV OC43 can be propagated in HCT-8 and Caco-2 (both are human epithelial colorectal adenocarcinoma cell lines) *Stanford University (2020)*, but neither of those cell types serves as a suitable monolayer for modified agarose-based plaque assay proposed in this study. Likewise, LLC-MK2 cells can generate more HCoV NL63 virus but are incompetent for plaque production of HCoV 229E (Fig. S3) despite both being alpha

coronaviruses (*Liu, Liang & Fung, 2020*). Nevertheless, Mv1Lu cells and the current improved agarose-based conventional plaque assay protocol are adequate for performing a low-cost consistent titration of HCoV OC43 and 229E. The development of this protocol is a significant step forward in the ability to titer HCoV strains and is a necessary step along the path of identifying effective antiviral agents against HCoV.

## SUPPLEMENTAL PROTOCOL

### Plaque assay (vero only)

A total of $5 \times 10^6$ Vero E6 cells were seeded in a 6-well plate for 24 h incubation at 33 °C. Prior to inoculation, the medium was removed and washed twice with PBS. Diluted viral samples were added to the cells for 1 h with rocking every 10 min. Upon removal of the inoculum cells were washed with PBS twice and 3 mL of overlay media, composed with MEM, 2% FBS, and 0.3% agarose, was added to each well. Plates were incubated at 33 °C for 3 days followed by fixation with 4% formaldehyde containing methanol at RT for 2 h. The plates were subsequently stained with a crystal violet solution.

## ACKNOWLEDGEMENTS

Human Coronavirus, Strain 229E (ATCC® VR-740™), FR-303 and Human Coronavirus, Strain OC43 (ATCC® VR-1558™), FR-302 were obtained through the International Reagent Resource, Influenza Division, WHO Collaborating Center for Surveillance, Epidemiology, and Control of Influenza, Centers for Disease Control and Prevention, Atlanta, GA, USA.

### Funding

The authors received no funding for this work.

### Competing Interests

The authors declare that they have no competing interests.

### Author Contributions

- Nicole Bracci conceived and designed the experiments, performed the experiments, analyzed the data, authored or reviewed drafts of the paper, and approved the final draft.
- Han-Chi Pan conceived and designed the experiments, analyzed the data, prepared figures and/or tables, authored or reviewed drafts of the paper, and approved the final draft.
- Caitlin Lehman performed the experiments, prepared figures and/or tables, and approved the final draft.
- Kylene Kehn-Hall analyzed the data, authored or reviewed drafts of the paper, and approved the final draft.
- Shih-Chao Lin conceived and designed the experiments, performed the experiments, analyzed the data, prepared figures and/or tables, authored or reviewed drafts of the paper, and approved the final draft.

PeerJ ___________________________________________________________________

## Data Availability

Raw data are available in the Supplemental Files.

## Supplemental Information

Supplemental information for this article can be found online at http://dx.doi.org/10.7717/peerj.10639#supplemental-information.

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
