# Peer review of "Improved plaque assay for human coronaviruses 229E and OC43"

_PeerJ, doi:10.7717/peerj.10639_

## Round 0.1 · original submission · Minor Revisions

Please provide a comprehensively revised version addressing the editorial comments and a detailed rebuttal letter

·

Basic reporting

1. The paper is lucid and I did not find any discernible mistakes in grammar or any typos in the spelling which tells of the hard work the authors have put in for preparing the manuscript. Kudos to the team for the presentation.

2. Bracci et al has eloquently described a protocol to perform plaque assay to screen antiviral compounds on non-severe coronavirus strains in a BSL-2 setting. The work is exhaustive given the global limitations imposed on research work in view of the pandemic and comes at an opportune time as well. The study has a solid and clear theoretical/conceptual framework, which makes it relevant to justify the study.

3. The purpose of the introduction is to orient the reader and create an interest in the study. This manuscript excels in that department without overstretching it.

Experimental design

1. The research question is clear. Furthermore, the rationale for the study is clear The authors have provided a clear statement about the need and relevance of this study.

2. The methods have been well described. Since this is primarily a methods paper, the Methods section should give readers enough information in such a form that they can repeat the experiments independently. In this manuscript, all methods are crystal clear. Thus, the authors deserve credit, since interested readers will surely find it easy to follow the same procedure.

Validity of the findings

1. The description of the results is also very well written. The data presented are easy to follow.
2. From my perspective, the weakest parts of the study are the images used. The images need to look at for brightness and contrast issues and may be due to the lack of photography skills of the experimenters but few images ESPECIALLY Fig 1 and Fig 7 do not feel professional at all. The authors could look into this aspect of an otherwise beautifully presented manuscript.

Additional comments

This is a very interesting study and kudos to the authors for sending such a manuscript
I would also like the authors to look for some professional help regarding the images for the correction of brightness and contrast.

Reviewer 2 ·

Basic reporting

The manuscript deals with the development of a new and improved plaque assay for titration of human coronaviruses 229E and OC43. A relevant method that may provide a better understanding about SARS-CoV-2. Very promising study.

Comments and suggestions:
1. An important advice for the authors is to consult a professional native English-speaking editing service for assistance with the manuscript, as there are several parts through it that should be carefully reviewed. I am attaching the reviewed PDF document with several recommendations that should be done to the manuscript.
2. From my perspective, the introduction should provide some further information to help readers to appreciate the relevance o the study. Thus, I would recommend adding a paragraph describing the rationale for using plaque assays to determine viral load, could help to improve the manuscript.
3. Even if scientific writing is not as stringent as before, in general terms it is highly recommended that scientific papers should be written in third person and using past tense, which implies avoiding the use of personal pronouns like "I", "we” or “our." So, several parts of the manuscript use these pronouns. Please, rewrite in a more academic style.
4. On lines 190 (page 11) and 263 (page 13) authors use the expression “we believe.” Science is not a matter of beliefs but of facts. Please, delete.
5. From my perspective, the study is very interesting and useful. I just would recommend adding a diagrammatic scheme resuming the more optimal conditions and media for titration of the human coronaviruses 229E and OC43. It may help readers interested in the standardization of this protocol at their laboratories.

Experimental design

No comments

Validity of the findings

The findings may be very relevant.

Additional comments

*** I UNDERSTAND THAT THE TASK OF THE REVIEWER IS TO SEE WHAT THE AUTHORS HAVE NOT SEEN. I THINK I AM PROVIDING AUTHORS WITH CONSTRUCTIVE CRITISISMS, AND I AM CERTAIN THAT THIS PROTOCOL COULD BE VERY RELEVANT FOR THE ADVANCE OF VIROLOGY. IF AUTHORS AGREE THAT THE MANUSCRIPT CAN BE IMPROVED BY CONSIDERING THE ABOVE MENTIONED, AND VERY SIMPLE, SUGGESTIONS, THE STUDY WOULD INCREASE ITS IMPACT.

Annotated reviews are not available for download in order to protect the identity of reviewers who chose to remain anonymous.

---

## Round 0.2 · accepted · Accept

Thanks for addressing the minor revisions requested. Now your manuscript is accepted in PeerJ.